# DECONFOUNDING TO EXPLANATION EVALUATION IN GRAPH NEURAL NETWORKS

## ABSTRACT

Explainability of graph neural networks (GNNs) aims to answer "Why the GNN made a certain prediction?", which is crucial to interpret the model prediction. The feature attribution framework distributes a GNN's prediction to its input features (*e.g.,* edges), identifying an influential subgraph as the explanation. When evaluating the explanation (*i.e.,* subgraph importance), a standard way is to audit the model prediction based on the subgraph solely. However, we argue that a distribution shift exists between the full graph and the subgraph, causing the *out-of-distribution* problem. Furthermore, with an in-depth causal analysis, we find the OOD effect acts as the confounder, which brings spurious associations between the subgraph importance and model prediction, making the evaluation less reliable. In this work, we propose Deconfounded Subgraph Evaluation (DSE) which assesses the *causal effect* of an explanatory subgraph on the model prediction. While the distribution shift is generally intractable, we employ the front-door adjustment and introduce a surrogate variable of the subgraphs. Specifically, we devise a generative model to generate the plausible surrogates that conform to the data distribution, thus approaching the unbiased estimation of subgraph importance. Empirical results demonstrate the effectiveness of DSE in terms of explanation fidelity.

## 1 INTRODUCTION

Explainability of graph neural networks (GNNs) (Hamilton et al., 2017; Dwivedi et al., 2020) is crucial to model understanding and reliability in real-world applications, especially when about fairness and privacy (Ying et al., 2019; Luo et al., 2020). It aims to provide insight into how **predictor** models work, answering "Why the target GNN made a certain prediction?". Towards this end, a variety of **explainer** models are proposed for feature attribution (Selvaraju et al., 2017; Ying et al., 2019; Luo et al., 2020; Vu & Thai, 2020), which decomposes the predictor's prediction as contributions (*i.e.,* importance) of its input features (*e.g.,* edges, nodes). While feature attribution assigns the features with importance scores, it redistributes the graph features and creates a new distribution different from that of the original full graphs, from which a subgraph is sampled as the explanation. Such sampling process is referred to as feature removal (Covert et al., 2020).

Then, to assess the explanatory subgraph, the current evaluation frameworks use the feature removal principle — (1) only feed the subgraph into the target predictor, discarding the other features; (2) measure the importance of the subgraph based on its information amount to recover the model's prediction. Such subgraph-prediction correlations uncovered by the removal-based **evaluator** should offer a faithful inspection of the predictor's decision-making process and assess the fidelity of the explainers reliably.

However, feature removal brings the out-of-distribution (OOD) problem (Frye et al., 2020; Chang et al., 2019; Lukas Faber, 2021): the distribution shift from full graphs to subgraphs likely violates underlying properties, including node degree distribution (Leskovec et al., 2005) and domain-specific constraints (Liu et al., 2018) of the full graphs. For example, graph properties of chemical molecules, such as the valency rules, impose some constraints on syntactically valid molecules (Liu et al., 2018); hence, simply removing some bonds (edges) or atoms (nodes) creates invalid molecular subgraphs that never appear in the training dataset. Such OOD subgraphs could manipulate the predictor's

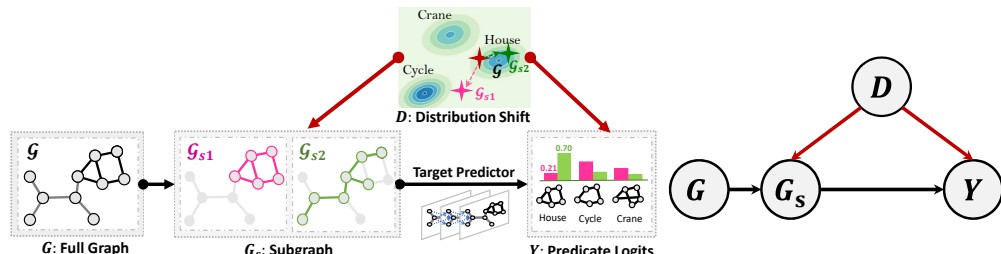

(a) Feature Removal to Evaluate Explanatory Subgraph $G_s$ (b) SCM I

Figure 1: (a) A real example in TR3. The GNN predictor classifies the full graph as 'House'. On subgraphs $\mathcal{G}_{s1}$ and $\mathcal{G}_{s2}$, the prediction probabilities of being "House" are respectively 0.21 and 0.70. (b) The structural causal model represents the causalities among variables: $G$ as the input graph, $D$ as the unobserved distribution shift, $G_s$ as the explanatory subgraph, and $Y$ as the model prediction.

outcome arbitrarily (Dai et al., 2018; Zügner et al., 2018), generates erroneous predictions, and limits the reliability of the evaluation process.

Here we demonstrate the OOD effect by a real example in Figure 1a, where the trained ASAP (Ranjan et al., 2020) predictor has classified the input graph as "House" for its attached motif (see Section 4 for more details). On the ground-truth explanation $\mathcal{G}_{s1}$, the output probability of the "House" class is surprisingly low (0.21). While for $\mathcal{G}_{s2}$ with less discriminative information, the outputs probability of the "House" class (0.70) is higher. Clearly, the removal-based evaluator assigns the OOD subgraphs with unreliable importance scores, which are unfaithful to the predictor's decision.

The OOD effect has not been explored in evaluating GNN explanations, to the best of our knowledge. We rigorously investigate it from a causal view (Pearl et al., 2016; Pearl, 2000; Pearl & Mackenzie, 2018). Figure 1b represents our causal assumption via a structural causal model (SCM) (Pearl et al., 2016; Pearl, 2000), where we target the causal effect of $G_s$ on $Y$. Nonetheless, as a **confounder** between $G_s$ and $Y$, distribution shift $D$ opens the spurious path $G_s \leftarrow D \rightarrow Y$. By "spurious", we mean that the path lies outside the direct causal path from $G_s$ to $Y$, making $G_s$ and $Y$ spuriously correlated and yielding an erroneous effect. And one can hardly distinguish between the spurious correlation and causative relations (Pearl et al., 2016). Hence, auditing $Y$ on $G_s$ suffers from the OOD effect and wrongly evaluates the importance of $G_s$.

Motivated by our causal insight, we propose a novel evaluation paradigm, **Deconfounded Subgraph Evaluator** (DSE), to faithfully measure the causal effect of explanatory subgraphs on the prediction. Based on Figure 1b, as the distribution shift $D$ is hardly measurable, we cannot block the backdoor path from $G_s$ to $Y$ by the backdoor adjustment. Thanks to the **front-door adjustment** (Pearl et al., 2016), we instead consider the SCM in Figure 2, where we introduce the surrogate $G_s^*$ between $G_s$ and $Y$, which "imagines" what the full graphs is like given the subgraphs. We obtain the causal effect of $G_s$ on $Y$ by identifying the causal effects carried by $G_s \rightarrow G_s^*$ and $G_s^* \rightarrow Y$, which requires $G_s^*$ to respect the data distribution. Hence we design a generative model, Conditional Variational Graph Auto-Encoder (CVGAE), to generate the possible surrogates. It is worthwhile mentioning that our DSE is explainer-agnostic, which can assist the explanation evaluation reliably and further guide explainers to generate faithful explanations.

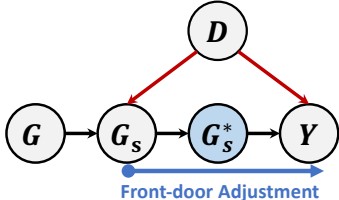

Figure 2: SCM II with a mediating variable $G_s^*$.

In a nutshell, our contributions are:

- From a causal perspective, we argue that the OOD effect is the confounder that causes spurious correlations between subgraph importance and model prediction.

- We propose a deconfounding paradigm, DSE, which exploits the front-door adjustment to mitigate the out-of-distribution effect and evaluate the explanatory subgraphs unbiasedly.

- We validate the effectiveness of our framework over various explainers, target GNN models, and datasets. Significant boosts are achieved over the conventional feature removal techniques. Code and datasets are available at: https://anonymous.4open.science/r/DSE-24BC/.

## 2 A CAUSAL VIEW OF EXPLANATION EVALUATION

Here we begin with the causality-based view of feature removal in Section 2.1 and present our causal assumption to inspect the OOD effect in Section 2.2.

### 2.1 PROBLEM FORMULATION

Without loss of generality, we focus on the graph classification task: a well-trained GNN predictor $f$ takes the graph variable $G$ as input and predicts the class $Y \in \{1, \cdots, K\}$, *i.e.*, $Y = f(G)$.

**Generation of Explanatory Subgraphs.** Post-hoc explainability typically considers the question "Why the GNN predictor $f$ made certain prediction?". A prevalent solution is building an explainer model to conduct feature attribution (Ying et al., 2019; Luo et al., 2020; Pope et al., 2019). It decomposes the prediction into the contributions of the input features, which redistributes the probability of features according to their importance and sample the salient features as an explanatory subgraph $\mathcal{G}_s$. Specifically, $\mathcal{G}_s$ can be a structure-wise (Ying et al., 2019; Luo et al., 2020) or feature-wise (Ying et al., 2019) subgraph of $\mathcal{G}$. In this paper, we focus on the structural features. That is, for graph $\mathcal{G} = (\mathcal{N}, \mathcal{E})$ with the edge set $\mathcal{E}$ and the node set $\mathcal{N}$, the explanatory subgraph $\mathcal{G}_s = (\mathcal{N}_s, \mathcal{E}_s)$ consists of a subset of edges $\mathcal{E}_s \subset \mathcal{E}$ and their endpoints $\mathcal{N}_s = \{u, v | (u, v) \in \mathcal{E}_s\}$.

**Evaluation of Explanatory Subgraphs.** Insertion-based evaluation by feature removal (Covert et al., 2020; Dabkowski & Gal, 2017) aims to check whether the subgraph is the supporting substructure [1] that alone allows a confident classification. We systematize this paradigm as three steps: (1) divide the full graph $\mathcal{G}$ into two parts, the subgraph $\mathcal{G}_s$ and the complement $\mathcal{G}_{\overline{s}}$; (2) feed $\mathcal{G}_s$ into the target GNN $f$, while discarding $\mathcal{G}_{\overline{s}}$; and (3) obtain the model prediction on $\mathcal{G}_s$, to assess its discriminative information to recover the prediction on $\mathcal{G}$. Briefly, at the core of the evaluator is the subgraph-prediction correlation. However, as discussed in Section 1, the OOD effect is inherent in the removal-based evaluator, hindering the subgraph-prediction correlation from accurately estimating the subgraph importance.

### 2.2 STRUCTURAL CAUSAL MODEL

To inspect the OOD effect rigorously, we take a causal look at the evaluation process with a Structural Causal Model (SCM I) in Figure 1b. We denote the abstract data variables by the nodes, where the directed links represent the causality. The SCM indicates how the variables interact with each other through the graphical definition of causation:

- $G \to G_s \leftarrow D$. We introduce an abstract distribution shift variable $D$ to sample a subgraph $G_s$ from the edge distributions of the full graph $G$.

- $G_s \to Y \leftarrow D$. We denote $Y$ as the prediction variable (*e.g.*, logits output), which is determined by (1) the direct effect from $G_s$, and (2) the confounding effect caused by $D$. In particular, the former causation that led to the result is the focus of this work.

We suggest readers to refer to Appendix A where we offer an elaboration of $D$. With our SCM assumption, directly measuring the importance of explanatory subgraphs is distracted by the backdoor path (Pearl, 2000), $G_s \leftarrow D \to Y$. This path introduces the confounding associations between $G_s$ and $Y$, which makes $G_s$ and $Y$ spuriously correlated, *i.e.*, biases the subgraph-prediction correlations, thus making the evaluator invalid. How to mitigate the OOD effect and quantify $G_s$'s genuine causal effect on $Y$ remains largely unexplored in the literature and is the focus of our work.

## 3 DECONFOUNDED EVALUATION OF EXPLANATORY SUBGRAPHS

In this section, we propose a novel deconfounding framework to evaluate the explanatory subgraphs in a trustworthy way. Specifically, we first leverage the front-door adjustment (Pearl, 2000) to formulate a causal objective in Section 3.1. We then devise a conditional variational graph auto-encoders (CVGAE) as the effective implementation of our objective in Section 3.2.

---

[1] We focus on insertion-based evaluation here while we discuss deletion-based evaluation in Appendix C.

### 3.1 FRONT-DOOR ADJUSTMENT

To the best of our knowledge, our work is the first to adopt the causal theory to solve the OOD problem in the explanation evaluation of GNNs. To pursue the causal effect of $G_s$ on $Y$, we perform the calculus of the causal intervention $P(Y = y|do(G_s = \mathcal{G}_s))$. Specifically, the *do-calculus* (Pearl, 2000; Pearl et al., 2016) is to intervene the subgraph variable $G_s$ by cutting off its coming links and assigning it with the certain value $\mathcal{G}_s$, making it unaffected from its causal parents $G$ and $D$. From inspection of the SCM in Figure 1b, the distribution effect $D$ acts as the confounder between $G_s$ and $Y$, and opens the backdoor path $G_s \leftarrow D \rightarrow Y$. However, as $D$ is hardly measurable, we can not use the backdoor adjustment (Pearl, 2000; Pearl et al., 2016) to block the backdoor path from $G_s$ to $Y$. Hence, the causal effect of $G_s$ on $Y$ is not identifiable from SCM I.

However, we can go much further by considering SCM II in Figure 2 instead, where a mediating variable $G_s^*$ is introduced between $G_s$ and $Y$:

- $G_s \rightarrow G_s^*$. $G_s^*$ is the surrogate variable of $G_s$, which completes $G_s$ to make them in the data distribution. First, it originates from and contains $G_s$. Specifically, it imagines how the possible full graphs should be when observing the subgraph $G_s$. Second, $G_s^*$ should follow the data distribution and respect the inherent knowledge of graph properties, thus no link exists between $D$ and $G_s^*$.
- $G_s^* \rightarrow Y$. This is based on our causal assumption that the causality-related information of $G_s$ on $Y$, *i.e.,* the discriminative information for $G_s$ to make prediction, is well-preserved by $G_s^*$. Thus, with the core of $G_s$, $G_s^*$ is qualified to serve as the mediator which further results in the model prediction.

With SCM II, we can exploit the front-door adjustment (Pearl, 2000; Pearl et al., 2016) instead to quantify the causal effect of $G_s$ on $Y$. Specifically, by summing over possible surrogate graphs $\mathcal{G}_s^*$ of $G_s^*$, we chain two identifiable partial effects of $G_s$ on $G_s^*$ and $G_s^*$ on $Y$ together:

$$
\begin{aligned}
P(Y|do(G_s = \mathcal{G}_s)) &= \sum_{\mathcal{G}_s^*} P(Y|do(G_s^* = \mathcal{G}_s^*))P(G_s^* = \mathcal{G}_s^*|do(G_s = \mathcal{G}_s)) \\
&= \sum_{\mathcal{G}_s^*} \sum_{\mathcal{G}_s'} P(Y|G_s^* = \mathcal{G}_s^*, G_s = \mathcal{G}_s')P(G_s = \mathcal{G}_s')P(G_s^* = \mathcal{G}_s^*|do(G_s = \mathcal{G}_s)) \\
&= \sum_{\mathcal{G}_s^*} \sum_{\mathcal{G}_s'} P(Y|G_s^* = \mathcal{G}_s^*, G_s = \mathcal{G}_s')P(G_s = \mathcal{G}_s')P(G_s^* = \mathcal{G}_s^*|G_s = \mathcal{G}_s), \quad (1)
\end{aligned}
$$

Specifically, we have $P(G_s^*|do(G_s = \mathcal{G}_s)) = P(G_s^*|G_s = \mathcal{G}_s)$ as $G_s$ is the only parent of $G_s^*$. And we distinguish the $\mathcal{G}_s$ in our target expression $P(Y|do(G_s = \mathcal{G}_s))$ between $\mathcal{G}_s'$, the latter of which is adjusted to pursue $P(Y|do(G_s^* = \mathcal{G}_s^*))$. With the data of $(\mathcal{G}_s, \mathcal{G}_s^*)$ pairs, we can obtain $P(Y|G_s^* = \mathcal{G}_s^*, G_s = \mathcal{G}_s')$ by feeding the surrogate graph $\mathcal{G}_s^*$ into the GNN predictor, conditional on the subgraph $\mathcal{G}_s'$; similarly, we can estimate $P(G_s = \mathcal{G}_s')$ statistically; $P(G_s^* = \mathcal{G}_s^*|G_s = \mathcal{G}_s)$ is the conditional distribution of the surrogate variable, after observing the subgraphs. As a result, this front-door adjustment yields a consistent estimation of $G_s$'s effect on $Y$ and avoids the confounding associations from the OOD effect.

### 3.2 DEEP GENERATIVE MODEL

However, it is non-trivial to instantiate $\mathcal{G}_s^*$ and collect the $(\mathcal{G}_s, \mathcal{G}_s^*)$ pairs. We get inspiration from the great success of generative models and devise a novel probabilistic model, conditional variational graph auto-encoder (CVGAE), and an adversarial training framework, to generate $\mathcal{G}_s^*$.

**Conditional Generation.** Inspired by previous works (Thomas N. Kipf, 2016; Liu et al., 2018), we model the data distribution via a generative model $g_\theta$ parameterized by $\theta$. It is composed of an encoder $q(\mathbf{Z}|\mathcal{G}, \mathcal{G}_s)$ and a decoder $p(\mathcal{G}_s^*|\mathbf{Z})$. Specifically, the encoder $q(\mathbf{Z}|\mathcal{G}, \mathcal{G}_s)$ embeds each node $i$ in $\mathcal{G}$ with a stochastic representation $\mathbf{z}_i$, and summarize all node representations in $\mathbf{Z}$:

$$
q(\mathbf{Z}|\mathcal{G}, \mathcal{G}_s) = \prod_{i=1}^{N} q(\mathbf{z}_i|\mathcal{G}, \mathcal{G}_s), \quad \text{with} \quad q(\mathbf{z}_i|\mathcal{G}, \mathcal{G}_s) = \mathcal{N}(\mathbf{z}_i \mid [\boldsymbol{\mu}_{1i}, \boldsymbol{\mu}_{2i}], \begin{bmatrix} \boldsymbol{\sigma}_{1i}^2 & 0 \\ 0 & \boldsymbol{\sigma}_{2i}^2 \end{bmatrix}) \quad (2)
$$

where $\mathbf{z}_i$ is sampled from a diagonal normal distribution by mean vector $[\boldsymbol{\mu}_{1i}, \boldsymbol{\mu}_{2i}]$ and standard deviation vector $\text{diag}(\boldsymbol{\sigma}_{1i}^2, \boldsymbol{\sigma}_{2i}^2)$; $\boldsymbol{\mu}_1 = f_\mu(\mathcal{G})$ and $\log \boldsymbol{\sigma}_1 = f_\sigma(\mathcal{G})$ denote the matrices of mean

Figure 3: Model structure of CVGAE. $P_{dse}$ is the average probability of $\mathcal{G}_s^*$ on the target prediction. AGG indicates the representations of the end nodes are aggregated as the edge embeddings.

vectors $\boldsymbol{\mu}_{1i}$ and standard deviation vectors $\log \boldsymbol{\sigma}_{1i}$ respectively, which are derived from two GNN models $f_\mu$ and $f_\sigma$ on the top of the full graph $\mathcal{G}$; similarly, $\boldsymbol{\mu}_2 = f_\mu(\mathcal{G}_s)$ and $\log \boldsymbol{\sigma}_2 = f_\sigma(\mathcal{G}_s)$ are on the top of the subgraph $\mathcal{G}_s$. Then, the decoder $p(\mathcal{G}_s^*|\mathbf{Z})$ generates the valid surrogates:

$$p(\mathcal{G}_s^*|\mathbf{Z}) = \prod_i^N \prod_j^N p(A_{ij}|\mathbf{z}_i, \mathbf{z}_j), \quad \text{with} \quad p(A_{ij} = 1|\mathbf{z}_i, \mathbf{z}_j) = f_A([\mathbf{z}_i, \mathbf{z}_j]), \tag{3}$$

where $A_{ij} = 1$ indicates the existence of an edge between nodes $i$ and $j$; $f_A$ is a MLP, which takes the concatenation of node representations $\mathbf{z}_i$ and $\mathbf{z}_j$ as the input and outputs the probability of $A_{ij} = 1$.

Leveraging the variational graph auto-encoder, we are able to generate some counterfactual edges that never appear in $\mathcal{G}$ and sample $\mathcal{G}_s^*$ from the conditional distribution $p(\mathcal{G}_s^*|\mathbf{Z})$, formally, $\mathcal{G}_s^* \sim p(G_s^*|\mathbf{Z})$. As a result, $P(G_s^* = \mathcal{G}_s^*|G_s = \mathcal{G}_s)$ in Equation 1 is identified by $p(\mathcal{G}_s^*|\mathbf{Z})$. The quality of the generator directly affects the quality of the surrogate graphs, further determines how well the front-door adjustment is conducted. Next, we will detail an adversarial training framework to optimize the generator, which is distinct from the standard training of VAE.

**Adversarial Training.** To achieve high-quality generation, we get inspiration from the adversarial training (Goodfellow et al., 2020; Yue et al., 2021) and devise the following training objective:

$$\min_\theta \mathcal{L}_{\text{VAE}} + \gamma \mathcal{L}_{\text{C}} + \max_\mu \omega \mathcal{L}_{\text{D}}, \tag{4}$$

where $\gamma, \omega$ are trade-off hyper-parameters. These losses are carefully designed to assure the generation follows the data distribution. Next, we will elaborate on each of them.

$$\mathcal{L}_{\text{VAE}} = -\mathbb{E}_\mathcal{G}[\mathbb{E}_{q(\mathbf{Z}|\mathcal{G}, \mathcal{G}_s)}[\log p(\hat{\mathcal{G}}_{\overline{s}}|\mathbf{Z})]] + \beta \mathbb{E}_\mathcal{G}[D_{\text{KL}}(q(\mathbf{Z}|\mathcal{G}, \mathcal{G}_s)||p(\mathbf{Z}))], \tag{5}$$

We first minimize the $\beta$-VAE loss(Higgins et al., 2017), and the first term is the reconstruction loss responsible to predict the probability of edges' existence; the second term is the KL-divergence between the variational and prior distributions. Here we resort to the isotropic Gaussian distribution $p(\mathbf{Z}) = \prod_i p(\mathbf{z}_i) = \prod_i \mathcal{N}(\mathbf{z}_i|\mathbf{0}, \mathbf{I})$ as the prior. $\beta$ reweighs the KL-divergence, which promises to learn the disentangled factors in $\mathbf{Z}$ (Higgins et al., 2017; Yue et al., 2021; Suter et al., 2019).

Moreover, we highlight the class-discriminative information in $\mathbf{Z}$, by encouraging the agreement between graph representations with the same class compared to that with different classes. Technically, the contrastive loss is adopted:

$$\mathcal{L}_{\text{C}} = -\mathbb{E}_\mathcal{G}[\log \frac{\sum_{\mathcal{G}' \in \mathcal{B}_+} \exp\left(s(\mathbf{z}_\mathcal{G}, \mathbf{z}_{\mathcal{G}'})/\tau\right)}{\sum_{\mathcal{G}'' \in \mathcal{B}_+ \cup \mathcal{B}_-} \exp\left(s(\mathbf{z}_\mathcal{G}, \mathbf{z}_{\mathcal{G}''})/\tau\right)}], \tag{6}$$

where $\mathbf{z}_\mathcal{G}$ is the representation of $\mathcal{G}$ that aggregates all node representations $\mathbf{Z}$ together; $s$ is the similarity function, which is given by an inner product here; $\tau$ is the temperature hyper-parameter; $\mathcal{B}_+$ is the graph set having the same class to $\mathcal{G}$, while the graphs involved in $\mathcal{B}_-$ have different classes from $\mathcal{G}$. Minimizing this loss enables the generator to go beyond the generic knowledge and uncover the class-wise patterns of graph data.

Besides, we introduce a discriminative model $d_\mu$ to distinguish the generated graphs. Specifically, we set it as a probability-conditional GNN (Fey & Lenssen, 2019) parameterized by $\mu$. It takes a graph as input and outputs a score between 0 to 1, which indicates the confidence of the graph being realistic. Hence, given a real graph $\mathcal{G}$ with the ground-truth label $y$, we can use the generator $g_\theta$ to generate $\mathcal{G}_s^*$. Then the discriminator learns to assign $\mathcal{G}$ with a large score while labeling $\mathcal{G}_s^*$ with a small score. To optimize the discriminator, we adopt the Wasserstein GAN (WGAN) (Martin Arjovsky, 2017) loss:

$$\mathcal{L}_{\text{D}} = \mathbb{E}_\mathcal{G}[\mathbb{E}_{p(\mathcal{G}_s^*|\mathbf{Z})}[d(\mathcal{G}, y) - d(\mathcal{G}_s^*, y) - \lambda(||\nabla_{\mathcal{G}_s^*} d(\mathcal{G}_s^*, y)||_2 - 1)^2]], \tag{7}$$

where $d(\mathcal{G}_s^*, y)$ is the probability of generating $\mathcal{G}_s^*$ from the generator; $\lambda$ is the hyper-parameter. By playing the min-max game between the generator and the discriminator in Equation 4, the generator can create the surrogate graphs from the data distribution plausibly.

**Subgraph Evaluation.** With the well-trained generator $g_\theta^*$ whose parameters are fixed, we now approximate the causal effect of $G_s$ on $Y$. Here we conduct Monte-Carlo simulation based on $g_\theta^*$ to sample a set of plausible surrogate graphs $\{\mathcal{G}_s^*\}$ from $p(\mathcal{G}_s^*|\mathbf{Z})$. Having collected the $(\mathcal{G}_s, \mathcal{G}_s^*)$ data, we can arrive the estimation of Equation 1.

## 4 EXPERIMENTS

We aim to answer the following research questions:

- **Study of Explanation Evaluation.** How effective is our DSE in mitigating the OOD effect and evaluating the explanatory subgraph more reliably? (Section 4.2)
- **Study of Generator.** How effective is our CVGAE in generating the surrogates for the explanatory subgraphs and making them conform to the data distribution? (Section 4.3)

### 4.1 EXPERIMENTAL SETTINGS

**Datasets & Target GNNs.** We first train various target GNN classifiers on the three datasets:

- **TR3** is a synthetic dataset involving 3000 graphs, each of which is constructed by connecting a random tree-shape base with one motif (house, cycle, crane). The motif type is the ground-truth label, while we treat the motifs as the ground-truth explanations following Ying et al. (2019); Yuan et al. (2020a). A Local Extremum GNN (Ranjan et al., 2019) is trained for classification.
- **MNIST superpixels (MNIST$_{\text{sup}}$)** (Monti et al., 2017) converts the MNIST images into 70,000 superpixel graphs. Every graph with 75 nodes is labeled as one of 10 classes. We train a Spline-based GNN (Fey et al., 2018) as the classifier model. The subgraphs representing digits can be viewed as human explanations.
- **Graph-SST2** (Yuan et al., 2020b) is based on text sentiment dataset SST2 (Socher et al., 2013) and converts the text sentences to graphs where nodes represent tokens and edges indicate relations between nodes. Each graph is labeled by its sentence sentiment. The node embeddings are initialized by the pre-trained BERT word embeddings (Devlin et al., 2018). Graph Attention Network (Veličković et al., 2018) is trained as the classifier.

**Ground-Truth Explanations.** By "ground-truth", we follow the prior studies (Ying et al., 2019; Yuan et al., 2020a; Luo et al., 2020) and treat the subgraphs coherent to the model knowledge (*e.g.,* the motif subgraphs in TR3) or human knowledge (*e.g.,* the digit subgraphs in MNIST$_{\text{sup}}$) as the ground-truth explanations. Although such ground-truth explanations might not fit the decision-making process of the model exactly, they contain sufficient discriminative information to help justify the explanations. Note that no ground-truth explanation is available in Graph-SST2.

**Explainers**. To explain the decisions made by these GNNs, we adopt several state-of-the-art explainers, including SA (Baldassarre & Azizpour, 2019), Grad-CAM (Selvaraju et al., 2017), GNNExplainer (Ying et al., 2019), CXPlain (Schwab & Karlen, 2019), PGM-Explainer (Vu & Thai, 2020), Screener (Anonymous, 2021), to generate the explanatory subgraphs. Specifically, top-15%, 20%, 20% of edges on the full graph instance construct the explanatory subgraphs in TR3, MNIST, and Graph-SST2, respectively. We refer readers to Appendix D for more experimental details.

### 4.2 STUDY OF EXPLANATION EVALUATION (RQ1)

**Deconfounded Evaluation Performance.** For an explanation $\mathcal{G}_s$, the conventional removal-based evaluation framework quantifies its importance as the subgraph-prediction correlation, termed $\text{Imp}_{\text{re}}(\mathcal{G}_s) = f(\mathcal{G}_s)$; whereas, our DSE framework focuses on the causal effect caused by $\mathcal{G}_s$ on $Y$ which is computed based on Equation 1, and we denote it as $\text{Imp}_{\text{dse}}(\mathcal{G}_s)$ for short. These importance scores broadly aim to reflect the discriminative information carried by $\mathcal{G}_s$. Thanks to the ground-truth knowledge available in TR3 and MNIST$_{\text{sup}}$, we are able to get a faithful and principled

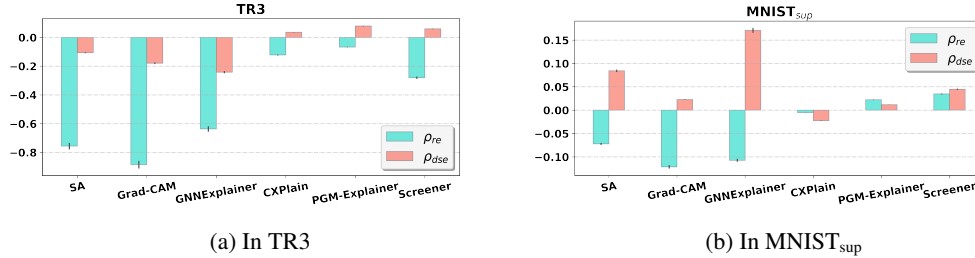

(a) In TR3                            (b) In MNIST$_{\text{sup}}$

Figure 4: Validation of different frameworks for explanation evaluation.

Table 1: Evaluation of explainers under different evaluation frameworks. $R_s$ is Spearman rank correlation function. Best explainers are underlined. Symbol $(\cdot)$ indicates the rank of explainers.

| | TR3 | | | MNIST$_{\text{sup}}$ | | | Graph-SST2 | | |
|---|---|---|---|---|---|---|---|---|---|
| | Imp$_{\text{re}}$(%) | Imp$_{\text{dse}}$(%) | Prec | Imp$_{\text{re}}$(%) | Imp$_{\text{dse}}$(%) | Prec | Imp$_{\text{re}}$(%) | Imp$_{\text{dse}}$(%) | Score |
| SA | | $43.23^{(1)}$ | $86.53^{(1)}$ | $17.60^{(3)}$ | $10.98^{(3)}$ | $32.98^{(2)}$ | $91.93^{(4)}$ | $95.67^{(4)}$ | $4.48^{(3)}$ |
| Grad-CAM | 33.07 | $43.18^{(2)}$ | $75.07^{(2)}$ | $16.90^{(5)}$ | $11.51^{(2)}$ | $31.42^{(3)}$ | $91.94^{(3)}$ | $96.21^{(2)}$ | $6.21^{(2)}$ |
| GNNExplainer | | $41.73^{(3)}$ | $56.34^{(4)}$ | $17.00^{(4)}$ | $12.27^{(1)}$ | $57.75^{(1)}$ | $89.40^{(5)}$ | $95.20^{(6)}$ | $4.26^{(4)}$ |
| CXPlain | | $38.61^{(6)}$ | $34.38^{(6)}$ | $14.30^{(6)}$ | $10.78^{(5)}$ | $11.14^{(5)}$ | $92.40^{(2)}$ | $95.98^{(3)}$ | $3.93^{(5)}$ |
| PGM-Explainer | 33.07 | $39.58^{(5)}$ | $48.47^{(5)}$ | $22.20^{(2)}$ | $10.77^{(6)}$ | $2.31^{(6)}$ | $89.16^{(6)}$ | $95.45^{(5)}$ | $1.68^{(6)}$ |
| Screener | | $40.31^{(4)}$ | $66.49^{(3)}$ | $32.20^{(1)}$ | $10.96^{(4)}$ | $19.51^{(4)}$ | $96.04^{(1)}$ | $96.39^{(1)}$ | $6.42^{(1)}$ |
| $R_s \uparrow$ | 0.011 | **0.943*** | - | $-0.142$ | **0.943*** | - | 0.657 | **0.714*** | - |

metric to measure the discriminative information amount — the precision $\text{Prec}(\mathcal{G}_s, \mathcal{G}_s^+)$ between the ground-truth explanation $\mathcal{G}_s^+$ and the explanatory subgraph $\mathcal{G}_s$. This precision metric allows us to perform a fair comparison between $\text{Imp}_{\text{re}}(\mathcal{G}_s)$ and $\text{Imp}_{\text{dse}}(\mathcal{G}_s)$ via:

$$\rho_{\text{re}} = \rho([\text{Prec}(\mathcal{G}_s, \mathcal{G}_s^+)], [\text{Imp}_{\text{re}}(\mathcal{G}_s)]), \quad \rho_{\text{dse}} = \rho([\text{Prec}(\mathcal{G}_s, \mathcal{G}_s^+)], [\text{Imp}_{\text{dse}}(\mathcal{G}_s)]), \quad (8)$$

where $\rho$ is the correlation coefficient between the lists of precision and importance scores. We present the results in Figure 4 and have some interesting insights:

- **Insight 1: Removal-based evaluation hardly reflects the importance of explanations.** In most cases, $\text{Prec}(\mathcal{G}_s, \mathcal{G}_s^+)$ is negatively correlated with the importance. This again shows that simply discarding a part of a graph could violate some underlying properties of graphs and mislead the target GNN, which is consistent with the adversarial attack works (Dai et al., 2018; Zügner et al., 2018). Moreover, the explainers that target high prediction accuracy, such as GNNExplainer, are easily distracted by the OOD effect and thus miss the important subgraphs.

- **Insight 2: Deconfounded evaluation quantifies the explanation importance more faithfully.** Substantially, $\rho_{\text{dse}}$ greatly improves after the frontdoor adjustments via the surrogate variable. The most notable case is GNNExplainer in MNIST$_{\text{sup}}$, where $\rho_{\text{dse}} = 0.17$ achieves a tremendous increase from $\rho_{\text{dse}} = -0.11$. Although our DSE alleviates the OOD problem significantly, weak positive or negative correlations still exist, which indicates the limitation of the current CVGAE. We leave the exploration of higher-quality generation in future work.

**Revisiting & Reranking Explainers.** Here we investigate the rankings of explainers generated from different evaluation frameworks, and further compute the Spearman rank correlations between these evaluation rankings and the reference rankings of explainers. Specifically, for TR3 and MNIST$_{\text{sup}}$ with ground-truth explanations, we regard the ranks *w.r.t.* precision as the references, while obtaining the reference of Graph-SST2 by a user study[2]. Such a reference offers the human knowledge for explanations and benchmarks the comparison. We show the results in Table 1 and conclude:

- **Insight 3: DSE presents a more fair and reliable comparison among explainers.** The DSE-based rankings are highly consistent with the references, while the removal-based rankings struggle to pass the check. In particular, we observe that for TR3, the unrealistic splicing inputs cause a plain ranking *w.r.t.* Imp$_{\text{re}}$. We find that various input subgraphs are predicted as cycle class. That is, the target GNN model is a deterministic gambler with serious OOD subgraphs. In contrast, DSE outputs a more informative ranking; For MNIST$_{\text{sup}}$, GNNExplainer with the highest precision

---

[2]70 volunteers are engaged, where each was asked to answer 10 questions randomly sampled from 32 movie reviews and choose the best explanations generated by the explainers. See Appendix E for more details.

Table 2: Importance scores or probabilities of subgraphs before and after feature removal.

|  | TR3 | MNIST$_{sup}$ | Graph-SST2 |
|---|---|---|---|
| Imp($G$) or GMM($G$) | $0.958_{-0.520}$ | $0.982_{-0.574}$ | $35.3_{-11.3}$ |
| Imp($G_s^+$) or GMM($G_s$) | 0.438 | 0.408 | 24.0 |

Table 3: Performances of Generators in terms of Validity and Fidelity.

|  | TR3 | | | MNIST$_{sup}$ | | | Graph-SST2 | | |
|---|---|---|---|---|---|---|---|---|---|
|  | Imp($G_s^*$) | VAL↑ | FID↓ | Imp($G_s^*$) | VAL↑ | FID↓ | GMM($G_s^*$) | VAL↑ | FID↓ |
| Random | 0.451 | 0.013 | 0.794 | 0.448 | 0.040 | 1.325 | 38.8 | 14.8 | 0.060 |
| VGAE | 0.469 | 0.031 | 0.754 | 0.205 | -0.203 | 1.501 | 37.6 | 13.6 | 0.078 |
| ARGVA | 0.392 | 0.061 | 0.726 | 0.466 | 0.058 | 1.306 | 31.0 | 7.0 | 0.079 |
| **CVGAE** | **0.603** | **0.165** | **0.598** | **0.552** | **0.144** | **0.910** | **45.8** | **21.8** | **0.057** |

is overly underrated by the removal-based evaluation framework, but DSE justifies its position faithfully; For Graph-SST2, although the OOD problem seems to be minor, DSE can still achieve significant improvement.

**Case Study.** We present a case study in Graph-SST2 to illustrate how DSE mitigates the potential OOD problem. See Appendix F for another case study on TR3. In Figure 5, $\mathcal{G}$ is a graph predicted as "negative" sentiment. The explanatory subgraph $\mathcal{G}_s$ emphasizes tokens like "weak" and relations like "n't→funny", which is cogent according to human knowledge. However, its removal-based importance is highly underestimated as $0.385$, possibly due to its disconnectivity or sparsity after feature removal. To mitigate the OOD problem, DSE samples $50$ surrogate graphs from the generator, performs the frontdoor adjustment, and justifies the subgraph importance as $0.913$, which shows the effectiveness of our DSE framework.

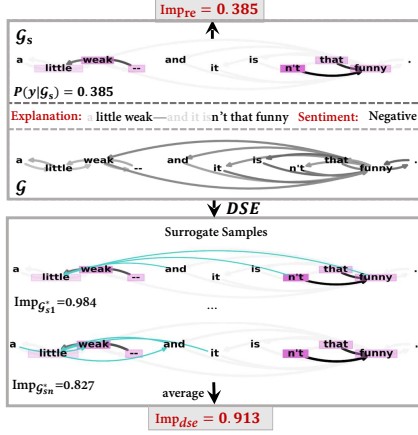

Figure 5: A Case Example.

We also observe some limitations of the generator (1) Due to the limited training data, the generators only reflect the distribution of the observed graphs, thus making some generations grammatically wrong. (2) The generations is constrained within the complete graph determined by the node set of the explanatory subgraph, thereby limits the quality of deconfounding. As we mainly focus on the OOD problem, we will leave the ability of the generator as future work.

## 4.3 STUDY OF GENERATORS (RQ2)

The generator plays an important role in our DSE framework, which aims to generate the valid surrogates conform to the data distribution. To evaluate the generator's quality, we compare it with three baselines: a random generator, a variational graph auto-encoder (VGAE) (Thomas N. Kipf, 2016), and an adversarially regularized variational graph auto-encoder (ARGVA) (Pan et al., 2018). We perform the evaluation based on two metrics: (1) **Validity.** For the ground-truth explanations $\mathcal{G}_s^+$ that contains all discriminative information of the full graph $\mathcal{G}$, the importance of its surrogate graph $\mathcal{G}_s^*$ should be higher than itself. The difference between the two importance scores indicates the validity of the generator, thus we define VAL $= \mathbb{E}_{\mathcal{G}}[\text{Imp}(\mathcal{G}_s^*) - \text{Imp}(\mathcal{G}_s^+)]$. For Graph-SST2 where the class-wise features are intractable, we leverage the embeddings of training graphs and additionally train a Gaussian Mixture Model (GMM) as our distribution prior. Then, we compute the average log-likelihood of random subgraphs after in-filling, thus we have VAL $= \mathbb{E}_{\mathcal{G}}\mathbb{E}_{\mathcal{G}_s \sim \text{Random}(\mathcal{G})}[\text{GMM}(\mathcal{G}_s^*) - \text{GMM}(\mathcal{G}_s)]$. (2) **Fidelity.** Towards a finer-grained assessment *w.r.t.* prediction probability of any random subgraphs, we adopt the metric following (Frye et al., 2021): FID $= \mathbb{E}_{\mathcal{G}}\mathbb{E}_{\mathcal{G}_s}\mathbb{E}_y|f_y(\mathcal{G}) - \mathbb{E}_{\mathcal{G}_s^*}[f_y(\mathcal{G}_s^*)]|^2$. This measures how well the surrogates cover the target prediction distribution.

Before comparing different generators, we first compute the importance or probabilities of the graphs before and after feature removal, which are summarized in Table 2. When inspecting the Removal's results without any in-fills, the OOD problem is severe: in TR3 and MNIST$_{sup}$, the importance of ground-truth subgraphs only reaches $43.8\%$ and $40.8\%$, respectively, which are far away from the

target importance of full graphs. Analogously in Graph-SST2. For the performance of the generators *w.r.t.* the two metrics, we summarize the average results over 5 runs in Table 3:

- The performance of the baselines are poor. This suggests that they can hardly fit the target conditional distribution.

- CVGAE outperforms other generators consistently across all cases, thus justifying the rationale and effectiveness of our proposed generator and adversarial training paradigm. For example, in TR3, CVGAE significantly increases the VAL scores and mitigates the OOD effect effectively.

Moreover, we conduct ablation studies and sensitivity analysis in Appendix G to better understand the model components and validate the effectiveness of the designed objective.

## 5 RELATED WORK

**Post-hoc Explainability of GNNs.** Inspired by the explainability in computer vision, Baldassarre & Azizpour (2019); Pope et al. (2019); Schnake et al. (2020) obtain the gradient-like scores of the model's outcome or loss *w.r.t.* the input features. Another line (Luo et al., 2020; Ying et al., 2019; Yuan et al., 2020a; Yue Zhang, 2020; Michael Sejr Schlichtkrull, 2021) learns the masks on graph features. Typically, GNN-Explainer (Ying et al., 2019) applies the instance-wise masks on the messages carried by graph structures, and maximizes the mutual information between the masked graph and the prediction. Going beyond the instance-wise explanation, PGExplainer (Luo et al., 2020) generates masks for multiple instances inductively. Recently, researchers adopt the causal explainability (Pearl & Mackenzie, 2018) to uncover the causation of the model predictions.For instance, CXPlain (Schwab & Karlen, 2019) quantifies a feature's importance by leaving it out. PGM-Explainer (Vu & Thai, 2020) performs perturbations on graph structures and builds an Bayesian network upon the perturbation-prediction pairs. Causal Screening (Screener) (Anonymous, 2021) measures the importance of an edge as its causal effect, conditional on the previously selected structures. Lately, SubgraphX (Yuan et al., 2021) explores different subgraphs with Monte-Carlo tree search and evaluates subgraphs with the Shapley value (Kuhn & Tucker, 1953).

**Counterfactual Generation for the OOD Problem.** The OOD effect of feature removal has been investigated in some other domains. There are generally two classes of generation (i) Static generation. For example, Fong & Vedaldi. (2017); Dabkowski & Gal (2017) adopted blurred input and random colors for the image reference, respectively. Due to the unnatural in-filling, the generated images are distributional irrespective and can still introduce confounding bias. (ii) Adaptive generation: Chang et al. (2019); Frye et al. (2021); Agarwal et al. (2019); Kim et al. (2020). The generators of these methods, like DSE, overcomes the defects aforementioned, which generates data that conforms to the training distribution. For example, in computer vision, FIDO (Chang et al., 2019) generates image-specific explanations that respect the data distribution, answering "Which region, when replaced by plausible alternative values, would maximally change classifier output?".

For the difference, firstly, DSE's formulated importance involves additional adjustment on $G_s$ and guarantees the unbiasedness of introducing the surrogate variable $G_s^*$, which is commonly discarded by the prior works with in-fillings only. Specifically, we offer a comparison with FIDO in Appendix B. Secondly, the distribution of graph data is more complicated to model than other domains. And the proposed CVGAE is carefully designed for graph data, where the contrastive loss and the adversarial training framework are shown to be effective for learning the data distribution of graphs.

## 6 CONCLUSION

In this work, we investigate the OOD effect on the explanation evaluation of GNNs. With a causal view, we uncover the OOD effect — the distribution shift between full graphs and subgraphs, as the confounder between the explanatory subgraphs and the model prediction, making the evaluation less reliable. To mitigate it, we propose a deconfounding evaluation framework that exploits the front-door adjustment to measure the causal effect of the explanatory subgraphs on the model prediction. And a deep generative model is devised to achieve the front-door adjustment by generating in-distribution surrogates of the subgraphs. In-so-doing, we can reliably evaluate the explanatory subgraphs. As the evaluation for explanations fundamentally guides the objective in GNNs explainability, this work offers in-depth insights into the future interpretability systems.

ETHICS STATEMENT

This work raises concerns about the removal-based evaluation in the explainability literature and proposed Deconfounded Subgraph Evaluator. For the user study that involves human subjects, we have detailed the fair evaluation procedure for each explanation generated by the explainers in Appendix E. For real-world applications, we admitted that the modeling of the distribution shift could be a barrier to fulfill their evaluation faithfulness. However, as shown in the paper, improper evaluation under the OOD setting largely biases the inspection of the model's decision-making process and the quality of explainers. Therefore, we argue that explainability should exhibit faithful explanation evaluation before auditing deep models' actual decision-making process. And a wrongly evaluated explanation might do more significant harm than an incorrect prediction, as the former could affect the general adjustment (*e.g.,* structure construction) and human perspective (*e.g.,* fairness check) of the model.

REPRODUCIBILITY STATEMENT

We have made great efforts to ensure reproducibility in this paper. Firstly, we make all causal assumptions clear in Section 2.2, Section 3.1 and Appendix A. For datasets, we have released the synthetic dataset, which can be referred to the link in Section 1, while the other two datasets are publicly available. We also include our code for model construction in the link. In Appendix D, we have reported the settings of hyper-parameters used in our implementation for model training.

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

# A    ELABORATION FOR OOD VARIABLE

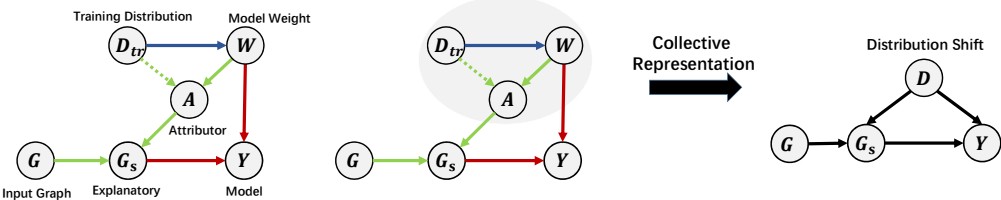

Figure 6: Elaboration for variable $D$ in Figure 1b, where the definitions of the variables are annotated. The blue, green, red arrows represent the data-generation process of model training, explaining, and removal-based evaluation, respectively; the dashed arrow $--\rightarrow$ denotes optional causal relation.

We offer a more fine-grained introduction of variable $D$ here. We first detail the data-generation process of model training, feature attribution and explanation evaluation of feature removal in Figure 6. Next, we justify each process:

- Model Training: $D_{tr} \rightarrow W$, which represents that the weights of the trained GNN $f$ is a random variable set by the optimizer, *e.g.,* a SGD, from a finite training dataset starting from random weights.

- Feature Attribution: Taking a view of certain explainers (Baldassarre & Azizpour, 2019; Selvaraju et al., 2017), the logits or probability vector from the last layer of trained GNN is redistributed using weights of the network onto the input and it will highlight the relevant edges. Thus, the causal relation $W \rightarrow A$ holds. Moreover, for parameterized explainers (Luo et al., 2020) which optimizes on the masks of the input features, the attributors (explainers) also results from the training distribution, forming the causal relation $D_{tr} \rightarrow A$. Finally, with input graphs $G$, we have $G \rightarrow G_s \leftarrow A$ where $G_s$ is the attribution from $A$'s algorithmic functions on the input graphs.

- Explanation Evaluation of Feature Removal: traditional evaluation methods simply evaluate the faithfulness of explanation via model forward, *i.e.,* $\hat{Y} = f_W(G_s)$ and regard the softmax readout on target prediction as the subgraph importance.

For simplicity, we combine variables $D_{tr}, W$ and $A$ collectively as an abstract distribution shift variable $D$ which is unobservable from real data. Thus, this justifies the existence of $D$ and our proposed SCM I.

# B    COMPARISON OF IMPORTANCE ESTIMATIONS

In this section, we compare our proposed estimation via front-door adjustment with the estimation in FIDO (Chang et al., 2019). We rephrased each estimation as

$$
\begin{aligned}
\text{Imp}_{dse}(\mathcal{G}_s) &= \sum_{\mathcal{G}_s^*} P\left(G_s^* = \mathcal{G}_s^* \mid G_s = \mathcal{G}_s\right) P\left(Y \mid G_s^* = \mathcal{G}_s^*\right) \\
&= \sum_{\mathcal{G}_s^*} P\left(G_s^* = \mathcal{G}_s^* \mid G_s = \mathcal{G}_s\right) \underbrace{\sum_{\mathcal{G}_s'} P\left(Y \mid G_s^* = \mathcal{G}_s^*, G_s = \mathcal{G}_s'\right) P\left(G_s = \mathcal{G}_s'\right)}
\end{aligned}
\tag{9}
$$

and

$$
\text{Imp}_{\text{FIDO}}(\mathcal{G}_s) = \sum_{\mathcal{G}_s^*} P\left(G_s^* = \mathcal{G}_s^* \mid G_s = \mathcal{G}_s\right) \underline{P\left(Y \mid G_s^* = \mathcal{G}_s^*\right)}
\tag{10}
$$

where DSE has alternatively adjusted on $G_s$ (represented as $G'_s$). To make it clear, we consider the underlined part of each equation. For Equation 9, we have

$$
\begin{aligned}
&\sum_{\mathcal{G}'_s} P\left(Y \mid G^*_s = \mathcal{G}^*_s, G_s = \mathcal{G}'_s\right) P\left(G_s = \mathcal{G}'_s\right) \\
&= \sum_{\mathcal{G}'_s} P\left(Y \mid G^*_s = \mathcal{G}^*_s, G_s = \mathcal{G}'_s\right) P\left(G_s = \mathcal{G}'_s \mid G^*_s = \mathcal{G}^*_s\right) \frac{P\left(G_s = \mathcal{G}'_s\right)}{P\left(G_s = \mathcal{G}'_s \mid G^*_s = \mathcal{G}^*_s\right)} \\
&= \sum_{\mathcal{G}'_s} P\left(Y, G_s = \mathcal{G}'_s \mid G^*_s = \mathcal{G}^*_s\right) \frac{P\left(G_s = \mathcal{G}'_s\right)}{\underline{P\left(G_s = \mathcal{G}'_s \mid G^*_s = \mathcal{G}^*_s\right)}}
\end{aligned}
\tag{11}
$$

While for the formulation of Equation 10, we have

$$
P\left(Y \mid G^*_s = \mathcal{G}^*_s\right) = \sum_{\mathcal{G}'_s} P\left(Y, G_s = \mathcal{G}'_s \mid G^*_s = \mathcal{G}^*_s\right)
\tag{12}
$$

In the comparison of these two parts, we can see that Equation 12 is biased under our causal assumption. Intuitively, each contribution of the importance of $\mathcal{G}^*_s$ on $Y$ should be inversely proportional to the posterior probability, *i.e.,* the probability of $\mathcal{G}'_s$ given the observation $\mathcal{G}^*_s$. However, FIDO fails to consider the causal relation between $G_s \to G^*_s$, which biases tha approximation of the genuine causal effect under our causal assumption.

Back to our proposed estimation, as we have collected $(\mathcal{G}_s, \mathcal{G}^*_s)$-pairs via Monte-Carlo simulation, thus additional adjustment on $\mathcal{G}_s$ $(\mathcal{G}'_s)$ can be achieved via Equation 11.

## C  DSE FOR DELETION-BASED EVALUATION

Based on the idea of deletion-based evaluation, we can instead use the average causal effect (Holland., 1988) (ACE) to look for a smallest deletion graph by conducting two interventions $do(G_s = \mathcal{G})$ (*i.e.,* , no feature removal) and $do(G_s = \mathcal{G}_{/s})$ where $\mathcal{G}_{/s}$ denotes the complement of the explanatory graph $\mathcal{G}_s$, meaning that the GNN input receives treatment and control, respectively. Formally, we have

$$
\mathrm{Imp}^{fid}_{dse}(G_s = \mathcal{G}_s) = P\left(Y \mid do\left(G_s = \mathcal{G}\right)\right) - P\left(Y \mid do\left(G_s = \mathcal{G}_{/s}\right)\right)
\tag{13}
$$

Then, we can similarly adjust for the individual terms as Equation 1, obtaining the unbiased importance value as the result of deletion-based evaluation.

## D  EXPERIMENTAL DETAILS

In this paper, all experiments are done on a single Tesla V100 SXM2 GPU (32 GB). The well-trained GNNs used in our experiments achieve high classification accuracies of 0.958 in TR3, 0.982 in $\mathrm{MNIST}_{\mathrm{sup}}$, 0.909 in Graph-SST2.

Now We introduce the model construction of the proposed generator. The encoder used is Crystal Graph Convolutional Neural Networks (Xie & Grossman, 2018), which contains three Convolutional layers. The encode dimensions in Tr3, $\mathrm{MNIST}_{\mathrm{sup}}$, Graph-SST2 datasets are respectively 256, 64, 256. For decoder, we adopt two fully connected layers with ReLU as activation layers, where the numbers of neurons are the same with the encode dimensions. Next, we summarize the pseudocodes for the Adversarial Training in Algorithm 1.

---

**Algorithm 1** Generative Adversarial Training. All experiments in the paper used the default values $m = 256$, $\alpha = 2 \times 10^{-4}$, $\beta = 1 \times 10^{-4}$, $\omega = \lambda = 5$, $\tau = 0.1$

---

**Require:** $\mathbb{P}_r$, real graphs' distribution. $r$, masking ratio.
**Require:** $m$, batch size. $\alpha$, learning rate. $\beta, \gamma, \lambda, \omega, \tau$, hyper-parameters.
1:  $\mu \leftarrow \mu_0$; $\theta \leftarrow \theta_0$
2:  **while** loss in Equation (4) is not converged **do**
3:      # Discriminator's training
4:      Sample $\{\mathcal{G}^{(i)}\}_{i=1}^{m} \sim \mathbb{P}_r$ a batch from the real graphs.
5:      Randomly generate broken graphs $\{\mathcal{G}_s^{(i)}\}_{i=1}^{m}$ from $\{\mathcal{G}^{(i)}\}_{i=1}^{m}$ with masking ratio $r$.
6:      Embed the nodes through encoder $q(\mathbf{Z}|\{\mathcal{G}_s^{(i)}, \mathcal{G}^{(i)}\}_{i=1}^{m})$
7:      Decode the edge probabilities and sample in-fill graphs $\{\hat{\mathcal{G}}_{\bar{s}}\}_{i=1}^{m} \sim p(\hat{G}_{\bar{s}} \mid \mathbf{Z})$
8:      Compute Discriminator's loss from Equation 7.
9:      Update parameter $\mu$ with back-propagation.
10:     # Generator's training
11:     Repeat the operations from line 4 to 7.
12:     Compute Generator's loss from Equation 4, 5, 6.
13:     Update parameter $\theta$ with back-propagation.
14: **end while**

---

For other hyper-parameters, we set $r = 0.3$, $\gamma = 3$ in Tr3 dataset. In MNIST$_{\text{sup}}$ and Graph-SST2 datasets, we set $r = 0.6$, $\gamma = 1$. We use Adam (Kingma & Ba, 2014) with weight decay rate 1e-5 for optimization. The maximum number of epochs is 100.

## E  DETAILED USER STUDY

The User Study starts by instructions to participants, where they will see a sentence (movie reviews) in each question and its sentiment (Positive of Negative), *e.g.,*

> Sentence: "is more of an ordeal than an amusement"
> Sentiment: Negative

then several explanations are presented for the answers of "Why the sentiment of this sentence is negative (positive)?". The explanations (see Figure 7) are shown in graph form (edges indicate relations between words), and colors of more important features are darker.

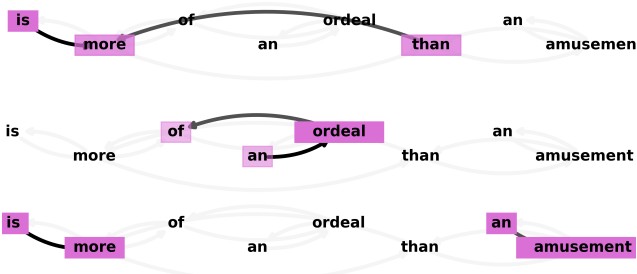

Figure 7: Instruction Example for conducting the user study.

Then they were asked to choose the best explanation(s). A good explanation should be concise, informative, and the rational cause of sentence's sentiment. In this case, (B) could be the best explanation since "ordeal" mostly decides the negative sentiment, while (A) only identifies plain words like "more than" and (C) is quite the opposite. Note that the participants can choose multiple answers and some choices are the same. Thereafter, 10 questions out of 32 questions in total are presented for each participant and we compute the average scores for the explainers.

## F    EXTRA CASE STUDY

In this section, we further present a case study for TR3 dataset. In Figure 8, the OOD probabilities for the ground truth explanatory subgraphs in each row remain the same as the edge selection ratios vary, which are 100%, 0%, 0% respectively. In contrast, the evaluation results generated from our DSE have shown strong rationality. Specifically, the importance score compute by our DSE increases with the increasing number of selected ground truth edges. This well validates our DSE framework, where we mitigate the OOD effect by generating the plausible surrogates, making the graphs to be evaluated conforms to the graph distribution in the training data. In this way, the effect of $D \rightarrow Y$ could hardly affect our assessment for the explanatory subgraph. Thereafter, as the explanatory graph becomes more informative and discriminative, it offers more evidence for the GNN to classify it as the target class which we want to explain, yielding faithful evaluation results.

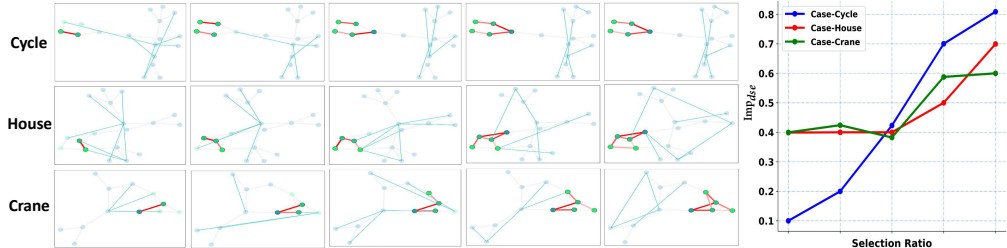

Figure 8: Three cases in TR3 datasets. Each graph in the left represents the ground truth explanatory subgraphs (red) for explaining a given graph. One of the complement graphs (light blue) generated from CVGAE is also shown with each explanatory subgraph. As the edge selection ratio increases in each row, the importance scores output by our DSE are shown in the right.

## G    ABLATION STUDY & SENSITIVITY ANALYSIS

We first conduct ablation studies to investigate the contribution of the contrastive parameter $\gamma$ and the penalty parameter $\lambda$ in CVGAE. The ablation models are proposed by I. removing the contrastive loss, *i.e.*, setting $\gamma = 0$ and II. removing the penalty term in the Wasserstein GAN (WGAN) (Martin Arjovsky, 2017) loss, *i.e.*, setting $\lambda = 0$. The performance of the ablation models is reported in Table 4. We observe that the superiority of CVGAE compared with the ablation model supports our model design by (i) smoothing the model optimization which yields a more performant generator (ii) highlighting the class-discriminative information in the graph embeddings, which implicitly encodes the class information.

Table 4: Ablation study on proposed CVGAE.

| Ablation Models | TR3 | | MNIST$_{sup}$ | | Graph-SST2 | |
|---|---|---|---|---|---|---|
| | VAL↑ | FID↓ | VAL↑ | FID↓ | VAL↑ | FID↓ |
| I. Remove $\mathcal{L}_C$ | 0.068 | 0.643 | 0.038 | 1.314 | 21.4 | 0.065 |
| II. Remove the penalty in $\mathcal{L}_D$ | 0.035 | 0.739 | 0.078 | 1.139 | 11.3 | 0.083 |
| **CVGAE** | **0.165**[*] | **0.598**[*] | **0.144**[*] | **0.910**[*] | **21.8**[*] | **0.057**[*] |

Also, we conduct sensitivity analysis for CVGAE *w.r.t.* the hyper-parameters. Specifically, we select $\lambda$, the penalty in the WGAN loss (*cf.* Euqation 7) and $\gamma$, the strength of the contrastive loss (*cf.* Equation 4). While we empirically found the performance is relatively indifferent to other parameters in a wide range. The results are shown in Figure 9. We observe that the best performance is achieved with $\lambda$ taking values from 1 to 10, and $\gamma$ taking values from 1 to 10 in TR3 dataset and 0.1 to 5 in MNIST$_{sup}$ and Graph-SST2 datasets. And we found a large $\lambda$ generally causes an increase in the FID metric, as it may alleviate the penalty on the reconstruction errors, which further makes a larger difference between $f_y(\mathcal{G})$ and $\mathbb{E}[f_y(\mathcal{G}_s^*)]$.

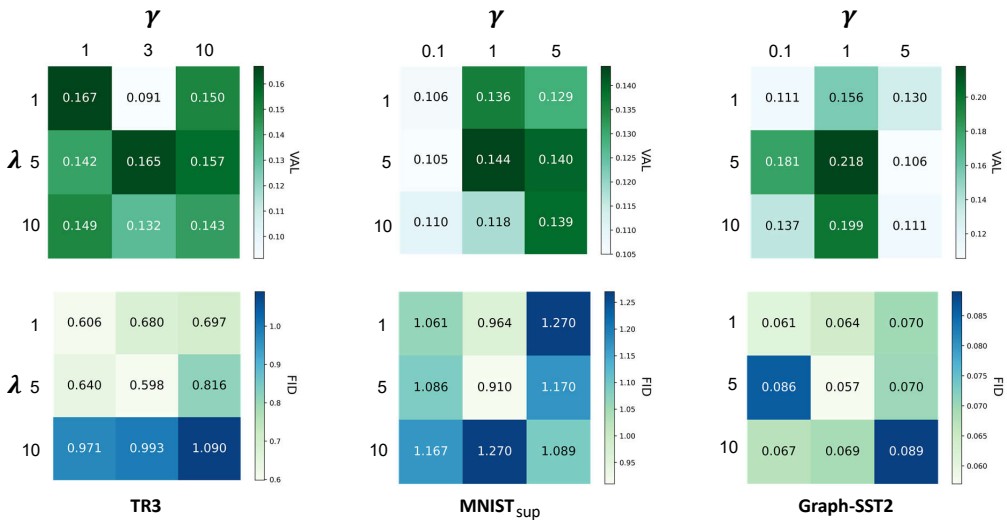

Figure 9: The performance of CVGAE using different $\lambda$ and $\gamma$ values.

