# OpenReview forum: "Deconfounding to Explanation Evaluation in Graph Neural Networks"
_ICLR.cc/2022/Conference — ICLR 2022 Submitted_

### Official Review · Reviewer_Tnp6 · 2021-10-16

**Correctness:** 3
**Technical Novelty And Significance:** 3
**Empirical Novelty And Significance:** 3
**Recommendation:** 8
**Confidence:** 4

**Main Review:**

Strengths:
In general, the motivation of this paper is very clear. The paper is easy to follow. View GNNs from a causal perspective is not a new idea. However, this paper uses causal theory to investigate the OOD problem in DNNs is new and interesting.

Weaknesses:

My major concerns are as follows,

1. In Section 3.1, the authors mention 'our work is the first to adopt the causal theory to \textbf{solve} the OOD problem ...'. However, I think this paper is a causal view on the OOD problem in DNNs instead of solving the OOD problem in DNNs. Especially, at the beginning of Section 4, the authors also highlight that they study the explanation evaluation and the generator, which are verified my opinion.

2. In the adversarial training part, there are several hyper-parameters such as $\gamma$, $\omega$, $\tau$, $\lambda$, and $\beta$. However, the authors do not provide any sensitivity analysis about these hyper-parameters. On the other hand, since there are three losses in Eq.(4), the authors should do several Ablation studies to demonstrate the significance of each loss function.

3. In Insight 3 of Section 4.2, the authors mention 'The DSE-based rankings are highly consistent with the references,'. However, it is not clear what is the references and where can we get these references? In specific, how to get the values of Prec column and values of Score column in Table 1.

4. Why the VAL performance of VGAE on MNIST$_{sup}$ is a negative value (-0.203)?

5. Since the work of Causal Screening (Screener) [1] is very close to this paper, the authors should discuss more differences between this paper and paper [1] instead of mentioning it slightly in the Related Work.

[1] Wang, Xiang, et al. "Causal Screening to Interpret Graph Neural Networks." (2020).

My minor concerns:

1. Some notations are not clear and there are some typos.
a) In Eq. (1), what is the meaning of \textbf{do}?
b) Below Eq.(3), 'Equation equation 1' should be 'Equation 1'.
c) In Figure 3, what is 'AGG'?

**Summary Of The Paper:**

In this paper, the authors use a causal view to investigate the OOD effect on the explanation evaluation of GNNs. They find the confounder between the extracted subgraphs and the model prediction, which makes the evaluation less reliable. To solve this problem, the authors proposed a deconfounding evaluation method based on the front-door adjustment from causal discovery. To generate a reliable surrogate subgraph, they proposed a generative model, which contains three losses for training. The experimental results show the effectiveness of the proposed method (DSE).

**Summary Of The Review:**

From my perspective, this paper is interesting in some aspects. The causal view on the OOD problem in DNNs is a new idea.

---

> ### Author Response · Authors · 2021-11-19
> **Response to Reviewer Tnp6**
>
> Thank you for the comments! Please find the answers to your concerns below.
>
> **Comment 1: Misused word.**
>
> This concern might be caused by the misunderstanding of our statement. Please see the complete sentence following Section 3.1, which states “our work is the first to adopt the causal theory to solve **the OOD problem in the explanation evaluation of GNNs**.” We refer the OOD problem to the biased assessment caused by feature removal, and we proposed DSE in Section 3 to solve it.
>
> **Comment 2: Sensitivity analysis & Ablation studies.**
>
> Please see Appendix G, where we have conducted ablation studies to investigate the contribution of each component loss, and we mentioned it in the paragraph above Section 5.
>
> For hyper-parameters, we have conducted model selection under the grid search. Following your advice, we conducted an additional sensitivity analysis, and we also included it in Appendix G. Please kindly see Appendix G, which is updated accordingly.
>
>
> **Comment 3: What are the references, and where can we get these references?**
>
> Please see the paragraph above insight 3, where we introduce the reference: “Speciﬁcally, for TR3 and MNIST with ground-truth explanations, we regard the ranks w.r.t. precision as the references, while obtaining the reference of Graph-SST2 by a user study.”
>
> You can also find the definition of **Prec** in the sentence below Table 1, which is defined as the precision between the ground-truth explanation $\mathcal{G}^+_s$ and the explanatory subgraph $\mathcal{G}_s$. And we also introduce how to get the reference rank from the user study in Appendix E, as indicated in the footnote.
>
> **Comment 4: Why the VAL performance of VGAE on MNIST is a negative value (-0.203)?**
>
> Good question. One reason is that the training of VGAE is not conditional, which makes it poor in generating the full graphs given the subgraph only. And we did look into the cases generated by VGAE in MNISTsup, and found the edges to be added are mostly within the subgraphs, i. e., the generated graphs are denser within the subgraphs but sparse outside the subgraphs. Thus, the generated graphs are further out-of-distribution and lose discriminative information within the ground truth subgraph $\mathcal{G}^+_s$, resulting in the negative VAL value. And this further demonstrates the superiority of our proposed CVAGE.
>
> **Comment 5: Similarity with Causal Screening (Screener) [1].**
>
> We respectfully argue our work has very little in common with [1]. The reasons are as follows.
>
> 1. Different tasks/scopes. Screener is an explainer model which tries to generate explanations for a trained model. Our proposed framework, DSE, is an evaluation framework that assesses the quality of explanations or explainers.
> 2. Different approaches. Although DSE and [1] both use causal reasoning, our work focuses on the OOD effect, which is mitigated by the front-door adjustment, while [1] computes the individual causal effect of an edge via simple intervention.
>
>
> **Notations.**
>
> 1. **"do"** means intervention which is commonly used in causal theory. For a variable $X$ and value $x$,  $do(X=x)$ means an intervention forces $X$ to attain fixed values $x$. Moreover, when $X$ is intervented, its parents (i.e., its causes in the causal graph) can no longer affect $X$. Thus, the causal intervention on $X$ defines a new distribution over the remaining variables that characterize the intervention's effect. Note $P(Y|do(X=x))$ is different from $P(Y|X=x)$, for the latter only reveals the probability observation while the former is more an experimental action. You can kindly refer to [2,3] if you need more details.
>
> 2.  **"AGG"** means aggregating the representations of end nodes, thus resulting in the concatenated edge embeddings. We explained the abbreviation in the caption of Figure 3 for better clarity.
>
> Typos. Thanks, we have fixed the typos.
>
>
> [1] Wang, Xiang, et al. "Causal Screening to Interpret Graph Neural Networks." (2020).
>
> [2] Judea Pearl and Dana Mackenzie.The book of why: the new science of cause and effect. Basic Books, 2018.
>
> [3] Judea Pearl, Madelyn Glymour, and Nicholas P Jewell. Causal inference in statistics: A primer.John Wiley & Sons, 2016.

---

### Official Review · Reviewer_pL1y · 2021-11-02

**Correctness:** 4
**Technical Novelty And Significance:** 4
**Empirical Novelty And Significance:** 3
**Recommendation:** 6
**Confidence:** 3

**Main Review:**

As I have stated in summary, the paper has caught an interesting problem with current explanation evaluation methods. This aspect is a strength! Another strength is that they cast the problem into a causal framework making the reasoning behind the novel evaluation mechanism reasonably interpretable.

The main issue with the paper lies in its clarity. It is not challenging to follow the technical details but rather to understand what the authors want to do. For instance, the authors title Section 2.1 as "Problem Formulation" but I do not see any problem formulated in that section.
Also, the evaluation method requires using a generative model to generate enough samples to be able to apply Equation 1.

For the above reasons, I would like authors to clearly state:
1. Whenever a new model for explaining GNNs is developed, should one also generate graphs using CVGAE?
2. How can one tell the superiority of the novel evaluation method from your paper? Where is exactly that assessed and proved? I tried to grasp it from reading the article, but I've not been able to.
3. Can you please clearly state what is the impact of the generative model on your evaluation? The two baselines are very weak, in my opinion. One is random, OK we should always include random baselines. The other one, though, is not conditional, which makes only explicit that a conditional generator is better. But I would have been surprised to find out that this was not the case. So, the question is: what are other baselines that would truly show the impact of the generator? Would it be sufficient to use whatever conditional method to generate graphs?


====== Minor Concerns ======
1. with less discriminative information --> Define "less discriminative information"
2. G_s is defined in the caption of Figure 1 but not in the text and when you first use it, is difficult to follow the sentence.
3. what the full graph like --> what the full graph is like
4. It harms the removal-based evaluation of the explanatory subgraph --> I've read this sentence over and over again but I've not been able to understand what you actually meant
5. well-trained GNN predictor --> What "well-trained" means?
6. it is rooted from G_s --> it is rooted on (?) G_s.  (What does actually mean this sentence?)
7. Equation (2) the product should have i = 1 and not just i
8. auto-encoder is able to generate --> autoencoder we are able to generate
9. the formula  that comes after formally does not look correct (or I've not understood it)
10. Besides, we introduce ... generated graphs --> Cannot understand it
11. Equation equation 1 (in a couple of places)



Authors have addressed most of my concerns, except for the #1, which is still not 100% clear. After the rebuttal, I've decided to raise my score.

**Summary Of The Paper:**

The paper sheds an exciting light on the problem of producing a meaningful evaluation of GNN explanation methods (at least a subset of them).
The idea is to introduce a deconfounder D to capture the effect of OOD explanations. The authors make an interesting example for a well-known synthetic dataset where the weight of the explanation in the ground truth is lower than a clear non-valid explanation when evaluated using the model to explain.
The introduction of the deconfounder D creates a spurious path between the graph variable and the explainer variable. To mitigate this effect, then, they introduce a front-door adjustment to the causal graph.
The front door adjustment requires a graph generator and authors use a novel Conditional-VGAE to generate graphs that will also cover the OOD case.
The paper finally presents some experiments showing the evaluation method in action.

**Summary Of The Review:**

In summary, I like the idea but I believe the paper does not make a good job of conveying the message to the readers.

---

> ### Author Response · Authors · 2021-11-19
> **Response to Reviewer pL1y**
>
> We appreciate your comments! To address your concerns, we provide point-to-point responses below.
>
> **Comment 1: Clarity for the title “Problem Formulation” in Section 2.1**
>
> This concern may be raised by the misunderstanding of our tasks or scopes. As our focus is on “evaluating” the “explanations” derived from the explainers, we first need to present the background of “generating explanation” and then touch on the core task of “evaluating explanation” in Section 2.1. Zooming in the part of “Evaluation of Explanatory Subgraphs”, we formulate the widely-used insertion-based evaluation framework and highlight the inherent OOD problem. The OOD problem is what we need to solve.
>
> **Comment 2: Should one also generate graphs using CVGAE when a new explainer is developed?**
>
> Our DSE (CVGAE) is independent of the explanation generation (i.e., developing a new explainer) but focuses on the explanation evaluation (i.e., evaluating the effectiveness of the developed explainer). Specifically, (1) it is explainer-agnostic and applicable to any explainer, having no influence on the explanation-generating process; (2) it helps us to reduce the OOD bias and fairly evaluate the quality of explanation, assessing which explainer is more reliable and powerful. Furthermore, it is more effective and promising in open-world scenarios, where the ground-truth explanations are unavailable.
>
> **Comment 3.1: The superiority of the novel evaluation method.**
>
> We first describe the superiority of our method in three folds:
> 1. **Theoretical Superiority.**
> From a causal perspective, we discovered a new finding that the OOD effect acts as the confounder that causes spurious correlations between subgraph importance and model prediction.
> 2. **Paradigm Superiority.** Based on the theoretical insights, we developed a novel deconfounding paradigm that uses the front-door adjustment to mitigate the bias and assess the subgraph fairly.
> 3. **Modeling Superiority.** We focus on a challenging data type, i.e., graph, and model the mapping from subgraph to full graph via the proposed CVGAE to realize the front-door adjustment, thus reliably evaluating the explanatory subgraphs.
>
>
> **Comment 3.2: How is the novel evaluation method assessed and proved?**
>
> 1. The theoretical and paradigm superiority is validated by causal theory.
> 2. We summarized the empirical effectiveness of the evaluation framework in the following table.
>
> |Assessment type|Location|Methods of assessment| Evidence of superiority |
> |:------:|:------|:-------|:-------|
> |Explainer-wise validation| Insight 1 & 2, Figure 4|We assess the evaluation framework for each explainer by checking the correlation coefficient between the lists of precision and evaluation scores, as defined in Eq. 8. We believe evaluation frameworks with higher correlation coefficients are more unbiased. |DSE significantly improves the correlation coefficient after the front-door adjustments via the surrogate variable.|
> |Model selection validation| Insight 3, Table 1|In each dataset, we compute spearman rank correlation between explainers rank from the evaluation framework and the ground truth rank, where we give priority to evaluation frameworks with higher rank correlation.|We see DSE generates rankings that are highly consistent with the references, while the removal-based ranks fail.|
> |Case-wise validation|Case studies in Section 4.2 and Appendix F|By examining the generated surrogate graphs, checking corrected importance, and observing how the importance value changes with precision.| The importance of explanations is justified by DSE. |
>
> We also separately validate the effectiveness of the generators, which is empirically proved in Section 4.3 under two designed metrics.

---

> > ### Author Response · Authors · 2021-11-19
> > **(cont.) Response to Reviewer pL1y**
> >
> >
> > **Comment 4.1: What is the impact of the generative model on the evaluation framework?**
> >
> > As pointed out in Section 4.3, the generators are used to generate valid surrogates that conform to the data distribution, which directly affects the approximation of Eq. 1.
> >
> > **Comment 4.2: What are other baselines that would truly show the impact of the generator?**
> >
> > Thanks. Firstly, we include another baseline ARGVA[1], which also exploits an adversarial training framework. We believe the comparison with a more advanced model can better show our model’s impact, which is summarized as follows.
> >
> > |||TR3|||MNIST$_{\text{sup}}$||| Graph-SST2| |
> > |:------:|:------|:-------|:-------|:------|:-------|:-------|:------|:-------|:-------|
> > ||Imp($G^∗_s$)|VAL$\uparrow$|FID$\downarrow$|Imp($G^∗_s$)|VAL$\uparrow$|FID$\downarrow$|GMM($G^∗_s$)|VAL$\uparrow$|FID$\downarrow$|
> > |ARGVA|$0.392$|$0.613$|$0.726$|$0.466$|$0.058$| $1.306$|$31.0$|$7.0$|$0.079$|
> > |**CVGAE** |$0.603$|$0.165$ | $0.598$ | $0.552$ |$0.144$|$0.910$|$45.8$|$21.8$|$0.057$|
> >
> > We can see CVGAE outperforms ARGVA by a large margin, which shows the advantage of our methods under the adversarial setting. The code is also updated accordingly.
> >
> > Secondly, the conditional generative model is one of our contributions, which is specially designed for the front-door adjustment. And it seems there is no well-developed conditional model for generating the full graphs, to the best of the authors’ knowledge.
> >
> >
> > **Typos & Sentence clarity.**
> >
> > Thank you. We have corrected the typos, defined $G_s$ in the text beside the caption in the figure, and rewrite the formula as $\mathcal{G}^*_s\sim p(G^*s|\mathbf{Z})$. Plus, we elaborate on the concerned terms/sentences:
> >
> > 1. **“Less discriminative information”** means given the subgraph pattern in $\mathcal{G}_{s2}$ (Figure 1), we are not able to recognize or make distinctions of it with accuracy.
> > 2. **“well-trained”** means the models have gained satisfactory training and reached high task performance. In our work, all of the trained GNNs have an accuracy higher than 95% (usually near 100%). And the generators are trained to gain a low validation loss.
> > 3. **“it is rooted from $G_s$”** means $G^*_s$ originates from and contains $G_s$. We have modified the sentence accordingly.
> > 4. **“We introduce a discriminative model $d_{\mu}$ to criticize the generated graphs”** means in the adversarial setting, the discriminative model $d_{\mu}$ is to distinguish whether the generated graphs are real or fake. We have changed the word “criticize” to “distinguish” for better clarity.
> >
> >
> > [1] Shirui Pan and Ruiqi Hu et al. Adversarially Regularized Graph Autoencoder for Graph Embedding. IJCAI, 2018.

---

> > > ### Author Response · Authors · 2021-11-23
> > > **Reminder**
> > >
> > > Dear Reviewer pL1y,
> > >
> > > As we are entering the final stage of discussion, it would be nice of you to let us know whether our answers have solved your concerns so that we can better improve our work. We are grateful for your time and efforts!

---

### Official Review · Reviewer_eQDt · 2021-11-02

**Correctness:** 3
**Technical Novelty And Significance:** 4
**Empirical Novelty And Significance:** 4
**Recommendation:** 8
**Confidence:** 4

**Main Review:**

The paper presents an interesting new method with a clear focus on debiasing GNN-explainer subgraph importance scores by generating surrogate subgraphs to correct distribution shift problems. The paper is well written and easy to follow, and the core idea sounds very effective and the empirical study using three datasets provides useful demonstrations.

Overall I liked the idea of the paper and found it nice work. Here are a couple of small questions to make sure of the paper's contribution.

- The proposed method is GNN-explainer agnostic. This point is advantageous because we can make importance corrections to any GNN explainers we like. But at the same time, a natural question will be: Is there any chance that we miss important features because the method to generate subgraph patterns didn't consider this OOD problem even if the proposed method can make a correction by posthoc processing. So for example, is it possible to have situations we have no significant important subgraph patterns after OOD-bias correction? Or, either way, any subgraph patterns come from at least one of the data graphs, and so by recovering such data graph within the training distribution, we can approximately resolve the data distribution bias and there are technically no problems??

- The paper focuses on structural features. This will imply that the bias under consideration is primarily distribution shift due to taking subgraphs of data graphs, and we need to recover the original data graph in the training set from a given any subgraph patterns.
But relatively small subgraph patterns can occur in multiple instances, and does this generative step actually generate such surrogate graph as intended?
Say, let G_s be a subgraph pattern we want to calculate the importance score but falls in out of the training distribution. The surrogate graph G_s^* by CVGAE is basically intended to recover the original super graph of G_s in the training dataset, isn't it? It'll be very helpful to make sure this point in some way.

- Just for your interest, and no need to include this in the paper: it might be interesting to consider whether the bias from subgraphs to the data graphs is the main problem. If the generative step can be explicit, we can even have a deterministic mapping from the possible subgraph patterns to the original graphs by graph mining algorithms. Explicit subgraph patterns are intensively investigated in the graph mining field in parallel to GNN-based approaches to the graph classification problem. For example, the book "Managing and Mining Graph Data (Ed: Charu C. Aggarwal & Haixun Wang) covers a list of relevant papers. So we can directly search explicit subgraph patterns in the data graphs and such approaches are intensively investigated around 10 years ago.
  - by direct graph mining such as mining "discriminative patterns", "emerging patterns", "contrastive patterns", (many works such as Llinares-López+, Fast and memory-efficient significant pattern mining via permutation testing, KDD2015)
  - by LASSO (K. Tsuda, Entire regularization paths for graph data. ICML 2007: 919-926)
  - by feature-wise boosting (Saigo+, gBoost: a mathematical programming approach to graph classification and regression, Machine Learning, 2009)
  - by sparse coordinate descent (Takigawa+, Generalized sparse learning of linear models over the complete subgraph feature set. TPAMI 2017)
  - by decision tree / decision forest (Shirakawa+, Jointly learning relevant subgraph patterns and nonlinear models of their indicators. MLG2018@KDD)
But using these methods, we often see disappointing conclusions that in general, "smaller subgraph patterns" are important because smaller subgraphs are more frequently occurring in the data and thus can be contributing in making predictions.


**Summary Of The Paper:**

This paper presents a novel explainer-agnostic method to adjust the biases of feature importance scores of feature attribution for GNNs. The paper first describes the feature importance scores of the GNN feature attribution framework have biases due to the out-of-distribution (OOD) problem. The subgraph important scores are calculated by inputting a subgraph instead of data graphs, but subgraph patterns can fall into regions outside the distribution of training data graphs. To address this problem, the paper proposed a method to generate surrogate graphs within the data graph distribution by CVGAE to make a front-door adjustment for deconfounding these biases by distribution shift. Experiments using several state-of-the-art GNN explainers shows demonstrated the effectiveness of the proposed framework.


**Summary Of The Review:**

The paper is well written, easy to follow, and presents a very useful explainer-agostic method to correct the OOD bias of subgraph important scores by GNN explainers. Small questions are made to make sure
1. Even this explainer-agnostic method can correct the OOD biases, the original GNN explainers didn't take into account those biases. Does this cause any problem?
2. What the surrogate graphs are like? They are recovered original graphs containing the given subgraphs?

---

> ### Author Response · Authors · 2021-11-19
> **Response to Reviewer eQDt**
>
> Thanks for your time and positive rating! To address your concerns, we provide point-to-point responses as follows.
>
> **Question 1.1: Could important features be missing due to the explainers’ neglect of the OOD problem?**
>
> Good catch! Yes, we think explainers could miss important features if distracted by the OOD effect. (This could also explain why some explainers are trained to a low loss but generate explanations with relatively low precisions.) In fact, building an explainer that considers the OOD effect is exactly what we want to do following this work, which goes beyond the post-hoc processing. Before that, we believe a well-rounded study of the OOD effect will be good for the research community, and a general method of correction will be beneficial for the existing explanatory methods and applications.
>
>
> **Question 2.1: What are the surrogate graphs like?**
>
> Please see the 2nd block of Figure 5, which shows the surrogate samples of the explanatory subgraph at the top of the figure. And yes, the surrogate samples contain the subgraph. But instead of the recovered original graph, they are a batch of graphs following the original distribution. For example, in TR3 dataset, given the subgraph of a $\textit{Tree-House}$ graph as the base graph $\textit{Tree}$, the surrogate samples could ideally contains $\textit{Tree-House}$, $\textit{Tree-Cycle}$ and $\textit{Tree-Crane}$ graphs, which are more than the original graph $\textit{Tree-House}$.
>
>
> **Question 2.2: Does the generative step actually generate such surrogate graphs as intended?**
>
> Yes. As stated above, the intended surrogate graphs are a batch of graphs containing the subgraph of interest, while following the original distribution. We empirically show that the proposed model can well generate surrogate graphs conforms to such data distribution in Section 4.3, and thus functions as intended to a certain extent.
>
> We also want to mention that it is hard to encode distributions of graph data, which partly contributes to the limitations of the generator, as suggested in the case study of Section 4.3. And we are intended to build a better model that encodes the graph distribution in our future work.
>
> **Question 1.2 Are there technical problems when resolving the data distribution bias if we have no significant important subgraph patterns?**
>
> This comment is insightful! We place the answer to this question after Q2.1 and Q2.2 for a better understanding. Again, we use the example when the explanatory subgraph of the $\textit{Tree-House}$ graph is $\textit{Tree}$, which has no discriminative subgraph pattern. In this case, the generative models are likely to generate various full graphs because the conditional probability of the full graphs given the subgraph is uniform in the three classes. Thus, such an uninformative subgraph won’t ideally get a score obviously over the score by random assignment, which validates our technical rationality.
>
>
> **Comment 3: Searching for explicit subgraph patterns.**
>
> Thanks! We have looked into a few works you mentioned, and these directions sound great. We believe it would be interesting, for example, to integrate graph mining algorithms, e.g., feature-wise boosting, with neural network-based mappings. Or using more structure level information from graph mining algorithms to assist the potential deterministic mapping in the embedding level. We will investigate more in future work. Thanks for the insightful suggestions again.

---

> > ### Comment · Reviewer_eQDt · 2021-11-22
> > **Thanks for your clarification!**
> >
> > Thank you for the answers! I acknowledge that I have read and understood the above responses. I have no further questions!

---

### Official Review · Reviewer_eWjy · 2021-11-03

**Correctness:** 4
**Technical Novelty And Significance:** 4
**Empirical Novelty And Significance:** 4
**Recommendation:** 8
**Confidence:** 3

**Main Review:**

This paper proposes a surrogate variable $G_{s}^*$ to denote the out-of-distribution effect and seems find interesting ways to evaluate the causal effects between the subgraph and full graph.

*Strengths*:

1. the out-of-distribution has not been explored before, as the paper claims.
2. the conditional variational graph autoencoder is well proposed and well trained.
3. I will the experimental settings, especially about the 3 insights. These results have sufficiently verify the claims and advantages of the paper.

*Weaknesses*
I did find clear weaknesses.

**Summary Of The Paper:**

This paper has done an excellent work of finding the out-of-distribution between the subgraph and graph as the confounder. Further, this paper proposes a conditional variational graph auto-encoder in assessing the causal effects of subgraph on the prediction. They also introduce a surrogate variable to denote this out-of-distribution effect. Through adversarial training, the effects of the proposed model is correctly verified.

**Summary Of The Review:**

Based on the innovation, clear model description and solid experimental results, I recommend for accept.

---

> ### Author Response · Authors · 2021-11-19
> **Response to Reviewer eWjy**
>
> Thank you for your comments! We are very grateful for your approval of our work.

---

### Author Response · Authors · 2021-11-19
**General Response**

We would like to thank all reviewers for their time and insightful suggestions. We are glad that most reviewers have positive ratings on our work. Here is a summary of our updates:

1. **More discussions**: We have more discussions about the influence of the OOD effect on explainers and the technical function of the generative model (eQDt). Also, we justify the statement made in the paper (Tnp6,pL1y), the problem formulation (pL1y), and highlight the difference between our work and the work of Screener (Tnp6).
2. **More experiments**: We add an adversarially regularized variational graph auto-encoder as an additional baseline, to show the impact of CVGAE (pL1y). Also, we conduct sensitivity analysis for a better understanding of our designed objective (Tnp6).

We hope our responses can clarify all your confusion and alleviate all concerns. We thank all reviewers again. We are looking forward to your reply!

---

### Decision · Program_Chairs · 2022-01-20

**Decision:**

Reject

**Comment:**

This paper recieves extensive discussions among SAC, two ACs and PCs. The decision was not made lightly. We hope that you will find the comments below  from two ACs useful for future publication.
----------------------------------------------------------------
This paper is concerned with the feature attribution framework, which distributes the prediction made by a Graph Neural Network (GNN) to its input features, such as edges or nodes, and identifies an influential subgraph as the explanation. The currently prevailing feature removal strategy, which feeds only the considered subgraph into the target predictor and then measures the importance of the subgraph, will encounter the so-called Out-Of-Distribution (OOD) problem--the new subgraphs may not appear in the training dataset.

This paper proposes to use the causal inference framework to deal with this OOD problem. The proposal seems interesting: it considered the considered subgraph as the cause of the prediction and treats domain shift as a (hidden) common cause for both of them. It proposes to estimate a surrogate graph G_s^* to satisfy the front-door criterion and then estimate the causal effect of the subgraph on the prediction.

While the proposed method seems novel and interesting and the reported empirical results seem encouraging, I have some basic concerns about the proposal.
1. The authors didn't justify why the feature attribution evaluation problem can/should be considered as a causal effect identification problem. In feature attribution evaluation, one is essentially interested in evaluating the prediction given the subgraph. The distribution shift variable, D, contains information that is helpful for this purpose.  Why should one go with the causal effect identification formulation, in which D is made independent from the subgraph and then integrated out? I think this justification is essential.

2. It is not justified why the constructed variable, G_s^*, satisfies the front-door criterion. In Section 3.1, the authors claimed that "G_s^* should follow the data distribution and respect the inherent knowledge of graph properties, thus no link exists between D and G_s^*."  I failed to see why this implies that there is no link from D to G_s^* (or that D and G_s^* are conditionally independent given G_s)--note that D is part of the data distribution. Without this justification, I am not sure whether the application of front-door adjustment is sensible.

Overall, the paper contains interesting ideas and the results look encouraging. It would be highly appreciated if the authors managed to make this work convincing, by properly addressing the issues above.  I hope to see those components in an updated paper.
------------------------------------------------------

This work considers the task of debiasing GNN-explainer subgraph importance scores by generating surrogate subgraphs to correct distribution shift problems. Through a causal model, the authors argue that a model prediction based on the explanatory subgraphs suffers from a distribution shift (e.g., induced subgraph degree distributions are different than those of the full graph). Hence, the associations between the important induced subgraphs and the model prediction may be spurious. In particular, the casting of induced subgraph explanations as a front-door hidden variable should be definitively valuable to the community (but maybe a little too strong of a condition, seems unnecessary for the proposed goal).

Overall the reviewers believe the goal and observations are novel and valuable. Most of the reviewers' concerns were addressed in the rebuttal. As a pure data-driven domain adaptation work, I think the work is good. However, even after discussing with the authors, I am still concerned with the soundness of the causal theory behind the work.

A Conditional Variational Graph Auto-Encoder (CVGAE) are tasked to produce subgraphs that act as a front-door adjustment variable. The argument is that this front-door adjustment solves the challenges of the distribution shift. Front-door adjustment is generally performed over observed variables. If performed with a model, the model is likely mechanistic. CVGAE are extremely flexible models for a front-door adjustment. And while the generated subgraphs are constrained to reproduce a data-driven distribution of induced subgraphs (and that those themselves have constraints), that in itself is not enough to guarantee these generated graphs give a proper front-door adjustment. The work also describes "generate counterfactual edges" without a clear causal model for how these edges are generated or why they are counterfactual. The causal theory needs a lot of work.

The work is very promising and may become a cornerstone contribution to graph explanations. However, as it stands now, the causal theory needs to be more formal (the work offers no proofs of the various claims). I am excited to see the causal theory fully developed in the future.